# DO NOT LET PRIVACY OVERBILL UTILITY: GRADIENT EMBEDDING PERTURBATION FOR PRIVATE LEARNING

**Da Yu**[1,2,*]**, Huishuai Zhang**[2,*]**, Wei Chen**[2]**, Tie-Yan Liu**[2]
[1]School of Computer Science and Engineering, Sun Yat-sen University
[2]Microsoft Research Asia
[1]`yuda3@mail2.sysu.edu.cn`
[2]`{huzhang,wche,tyliu}@microsoft.com`

## ABSTRACT

The privacy leakage of the model about the training data can be bounded in the differential privacy mechanism. However, for meaningful privacy parameters, a differentially private model degrades the utility drastically when the model comprises a large number of trainable parameters. In this paper, we propose an algorithm *Gradient Embedding Perturbation (GEP)* towards training differentially private deep models with decent accuracy. Specifically, in each gradient descent step, GEP first projects individual private gradient into a non-sensitive anchor subspace, producing a low-dimensional gradient embedding and a small-norm residual gradient. Then, GEP perturbs the low-dimensional embedding and the residual gradient separately according to the privacy budget. Such a decomposition permits a small perturbation variance, which greatly helps to break the dimensional barrier of private learning. With GEP, we achieve decent accuracy with reasonable computational cost and modest privacy guarantee for deep models. Especially, with privacy bound $\epsilon = 8$, we achieve $74.9\%$ test accuracy on CIFAR10 and $95.1\%$ test accuracy on SVHN, significantly improving over existing results.

## 1 INTRODUCTION

Recent works have shown that the trained model may leak/memorize the information of its training set (Fredrikson et al., 2015; Wu et al., 2016; Shokri et al., 2017; Hitaj et al., 2017), which raises privacy issue when the models are trained with sensitive data. *Differential privacy* (DP) mechanism provides a way to quantitatively measure and upper bound such information leakage. It theoretically ensures that the influence of any individual sample is negligible with the DP parameter $\epsilon$ or $(\epsilon, \delta)$. Moreover, it has been observed that differentially private models can also resist model inversion attack (Carlini et al., 2019), membership inference attack (Rahman et al., 2018; Bernau et al., 2019; Sablayrolles et al., 2019; Yu et al., 2021), gradient matching attack (Zhu et al., 2019), and data poisoning attack (Ma et al., 2019).

One popular way to achieve differentially private machine learning is to perturb the training process with noise (Song et al., 2013; Bassily et al., 2014; Shokri & Shmatikov, 2015; Wu et al., 2017; Fukuchi et al., 2017; Iyengar et al., 2019; Phan et al., 2020). Specifically, *gradient perturbation* perturbs the gradient at each iteration of (stochastic) gradient descent algorithm and guarantees the privacy of the final model via *composition property* of DP. It is worthy to note that gradient perturbation does not assume (strongly) convex objective and hence is applicable to various settings (Abadi et al., 2016; Wang et al., 2017; Lee & Kifer, 2018; Jayaraman et al., 2018; Wang & Gu, 2019; Yu et al., 2020). Specifically, for given gradient sensitivity $S$, a general form of gradient perturbation is to add an isotropic Gaussian noise $\boldsymbol{z}$ to the gradient $\boldsymbol{g} \in \mathbb{R}^p$ independently for each step,

$$\tilde{\boldsymbol{g}} = \boldsymbol{g} + \boldsymbol{z}, \ \text{ where } \ \boldsymbol{z} \sim \mathcal{N}(0, \sigma^2 S^2 \boldsymbol{I}_{p \times p}). \tag{1}$$

One can set proper variance $\sigma^2$ to make each update differentially private with parameter $(\epsilon, \delta)$. It is easy to see that the intensity of the added noise $\mathbb{E}[\|\boldsymbol{z}\|^2]$ scales linearly with the model dimension $p$.

---

*Authors contribute equally to this work.

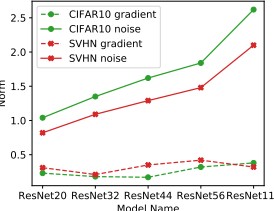

Figure 1: Noise norm vs gradient norm of ResNet20 at initialization. The noise variance is chosen such that SGD satisfies $(5, 10^{-5})$-DP after 90 epochs in Abadi et al. (2016).

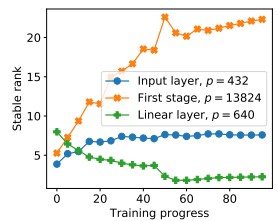

Figure 2: Stable rank $\|\cdot\|_F^2/\|\cdot\|^2$ (Tropp et al., 2015) of batch gradient matrix of given groups (with $p$ parameters). The setting is ResNet20 on CIFAR-10. The stable rank is small throughout training.

This indicates that as the model becomes larger, the useful signal, i.e., gradient, would be submerged in the added noise (see Figure 1). This dimensional barrier restricts the utility of deep learning models trained with gradient perturbation.

The dimensional barrier is attributed to the fact that the added noise is isotropic while the gradients live on a very low dimensional manifold, which has been observed in (Gur-Ari et al., 2018; Vogels et al., 2019; Gooneratne et al., 2020; Li et al., 2020) and is also verified in Figure 2 for the gradients of a 20-layer ResNet (He et al., 2016). Hence to limit the noise energy, it is natural to think

"Can we reduce the dimension of gradients first and then add the isotropic noise onto a low-dimensional gradient embedding?"

The answer is affirmative. We propose a new algorithm *Gradient Embedding Perturbation (GEP)*, illustrated in Figure 3. Specifically, we first compute *anchor gradients* on some non-sensitive auxiliary data, and identify an *anchor subspace* that is spanned by several top principal components of the anchor gradient matrix. Then we project the private gradients into the anchor subspace and obtain low-dimensional *gradient embeddings* and small-norm *residual gradients*. Finally, we perturb the gradient embedding and residual gradient separately according to the sensitivities and privacy budget.

We intuitively argue why GEP could reduce the perturbation variance and achieve good utility for large models. First, because the gradient embedding has a very low dimension, the added isotropic noise on embedding has small energy that scales linearly only with the subspace dimension. Second, if the anchor subspace can cover most of the gradient information, the residual gradient, though high dimensional, should have small magnitude, which permits smaller added noise to guarantee the same level privacy because of the reduced sensitivity. Overall, we can use a much lower perturbation compared with the original gradient perturbation to guarantee the same level of privacy.

We emphasize several properties of GEP. First, the non-sensitive auxiliary data assumption is weak. In fact, GEP only requires a small number of non-sensitive unlabeled data following a similar feature distribution as the private data, which often exist even for learning on sensitive data. In our experiments, we use a few unlabeled samples from ImageNet to serve as auxiliary data for MNIST, SVHN, and CIFAR-10. This assumption is much weaker than the public data assumption in previous works (Papernot et al., 2017; 2018; Alon et al., 2019; Wang & Zhou, 2020), where the public data should follow exactly the same distribution as the private data. Second, GEP produces an unbiased estimator of the target gradient because of releasing both the perturbed gradient embedding and the perturbed residual gradient, which turns out to be critical for good utility. Third, we use *power method* to estimate the principal components of anchor gradients, achievable with a few matrix multiplications. The fact that GEP is not sensitive to the choices of subspace dimension further allows a very efficient implementation.

Compared with existing works of differentially private machine learning, our contribution can be summarized as follows: (1) we propose a novel algorithm GEP that achieves good utility for large models with modest differential privacy guarantee; (2) we show that GEP returns an unbiased estimator of target private gradient with much lower perturbation variance than original gradient perturbation; (3) we demonstrate that GEP achieves state-of-the-art utility in differentially private learning with three benchmark datasets. Specifically, for $\epsilon = 8$, GEP achieves $74.9\%$ test accuracy

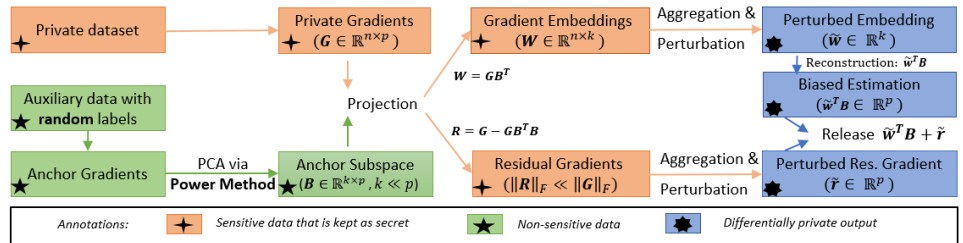

Figure 3: Overview of the proposed GEP approach. 1) We estimate an anchor subspace on some non-sensitive data; 2) We project the private gradients into the anchor subspace, producing low-dimensional embeddings and residual gradients; 3) We perturb the gradient embedding and residual gradient separately to guarantee differential privacy. The auxiliary data are only required to share similar features as the private data. In our experiments, we use 2000 images from ImageNet as auxiliary data for MNIST, SVHN, and CIFAR-10 datasets.

on CIFAR-10 with a ResNet20 model. To the best of our knowledge, GEP is the first algorithm that can achieve such utility with training deep models from scratch for a "single-digit" privacy budget[1].

## 1.1 RELATED WORK

Existing works studying differentially private machine learning in high-dimensional setting can be roughly categorized into two sets. One is treating the optimization of the machine learning objective as a whole mechanism and adding noise into this process. The other one is based on the knowledge transfer of machine learning models, which trains a differentially private publishable student model with private signals from teacher models. We review them one by one.

Differentially private convex optimization in high-dimensional setting has been studied extensively over the years (Kifer et al., 2012; Thakurta & Smith, 2013; Talwar et al., 2015; Wang & Xu, 2019; Wang & Gu, 2019). Although these methods demonstrate good utility on some convex settings, their analyses can not be directly applied to non-convex setting. Right before the submission, we note two independent and concurrent works (Zhou et al., 2020; Kairouz et al., 2020) that also leverage the gradient redundancy to reduce the added noise. Specifically, Kairouz et al. (2020) track historical gradients to do dimension reduction for private AdaGrad. Zhou et al. (2020) requires gradients on some public data and then project the noisy gradients into a public subspace at each update. One core difference between these two works and GEP is that we introduce residual gradient perturbation and GEP produces an unbiased estimator of the private gradients, which is essential for achieving the superior utility. Moreover, we weaken the auxiliary data assumption and introduce several designs that significantly boost the efficiency and applicability of GEP.

One recent progress towards training arbitrary models with differential privacy is *Private Aggregation of Teacher Ensembles (PATE)* (Papernot et al., 2017; 2018; Jordon et al., 2019). PATE first trains independent teacher models on disjoint shards of private data. Then it trains a student model with privacy guarantee by distilling noisy predictions of teacher models on some public samples. In comparison, GEP only requires some non-sensitive data that have similar natural features as the private data while PATE requires the public data follow exactly the same distribution as the private data and in practice it uses a portion of the test data to serve as public data. Moreover, GEP demonstrates better performance than PATE especially for complex datasets, e.g., CIFAR-10, because GEP can train the model with the whole private data rather than a small shard of data.

## 2 PRELIMINARIES

We introduce some notations and definitions. We use bold lowercase letters, e.g., $v$, and bold capital letters, e.g., $M$, to denote vectors and matrices, respectively. The $L^2$ norm of a vector $v$ is denoted by $\|v\|$. The spectral norm and the Frobenius norm of a matrix $M$ are denoted by $\|M\|$ and $\|M\|_F$, respectively. A sample $d = (x, y)$ consists of feature $x$ and label $y$. A dataset $\mathbb{D}$ is a collection of individual samples. A dataset $\mathbb{D}'$ is said to be a neighboring dataset of $\mathbb{D}$ if they differ in a

---

[1] Abadi et al. (2016) achieve 73% accuracy on CIFAR-10 but they need to pre-train the model on CIFAR-100.

single sample, denoted as $\mathbb{D} \sim \mathbb{D}'$. Differential privacy ensures that the outputs of an algorithm on neighboring datasets have approximately indistinguishable distributions.

**Definition 1** (($\epsilon, \delta$)-DP (Dwork et al., 2006a;b))**.** *A randomized mechanism $\mathcal{M}$ guarantees ($\epsilon, \delta$)-differential privacy if for any two neighboring input datasets $\mathbb{D} \sim \mathbb{D}'$ and for any subset of outputs $S$ it holds that $Pr[\mathcal{M}(\mathbb{D}) \in S] \leq e^\epsilon Pr[\mathcal{M}(\mathbb{D}') \in S] + \delta$.*

By its definition, ($\epsilon, \delta$)-DP controls the maximum influence that any individual sample can produce. One can adjust the privacy parameters to trade off between privacy and utility. Differential privacy is immune to *post-processing* (Dwork et al., 2014), i.e., any function applied on the output of a differentially private algorithm would not increase the privacy loss as long as it does not have new interaction with the private dataset. Differential privacy also allows *composition*, i.e., the composition of a series of differentially private mechanisms is also differentially private but with different parameters. Several variants of ($\epsilon, \delta$)-DP have been proposed (Bun & Steinke, 2016; Dong et al., 2019) to address certain weakness of ($\epsilon, \delta$)-DP, e.g., they achieve better composition property. In this work, we use *Rényi differential privacy* (Mironov, 2017) to track the privacy loss and then convert it to ($\epsilon, \delta$)-DP.

Suppose that there is a private dataset $\mathbb{D} = \{(\boldsymbol{x}_i, y_i)\}_{i=1}^n$ with $n$ samples. We want to train a model $f$ to learn the mapping in $\mathbb{D}$. Specifically, $f$ takes $\boldsymbol{x}$ as input and outputs a label $y$, and $f$ has parameter $\theta \in \mathbb{R}^p$. The training objective is to minimize an empirical risk $\frac{1}{n} \sum_{i=1}^n \ell(f(\boldsymbol{x}_i), y_i)$, where $\ell(\cdot, \cdot)$ is a loss function. We further assume that there is an auxiliary dataset $\mathbb{D}^{(a)} = \{(\tilde{\boldsymbol{x}}_j, \tilde{y}_j)\}_{j=1}^m$ that $\tilde{\boldsymbol{x}}$ shares similar features as $\boldsymbol{x}$ in $\mathbb{D}$ while $\tilde{y}$ could be random.

## 3 Gradient embedding perturbation

An overview of GEP is given in Figure 3. GEP has three major ingredients: 1) first, estimate an anchor subspace that contains the principal components of some non-sensitive anchor gradients via power method; 2) then, project private gradients into the anchor subspace and produce low-dimensional embeddings of private gradients and residual gradients; 3) finally, perturb gradient embedding and residual gradient separately to establish differential privacy guarantee. In Section 3.1, we present the GEP algorithm in detail. In Section 3.2, we given an analysis on the residual gradients. In Section 3.3, we give a differentially private learning algorithm that updates the model with the output of GEP.

### 3.1 The GEP algorithm and its privacy analysis

The pseudocode of GEP is presented in Algorithm 1. For convenience, we write a set of gradients and a set of basis vectors as matrices with each row being one gradient/basis vector.

The anchor subspace is constructed as follows. We first compute the gradients of the model on an auxiliary dataset $\mathbb{D}^{(a)}$ with $m$ samples, which is referred to as the anchor gradients $\boldsymbol{G}^{(a)} \in \mathbb{R}^{m \times p}$. We then use the power method to estimate the principal components of $\boldsymbol{G}^{(a)}$ to construct a subspace basis $\boldsymbol{B} \in \mathbb{R}^{k \times p}$, which is referred to as the anchor subspace. All these matrices are publishable because $\mathbb{D}^{(a)}$ is non-sensitive. We expect that the anchor subspace $\boldsymbol{B}$ can cover most energy of private gradients when the auxiliary data are not far from private data and $m, k$ are reasonably large.

Suppose that the private gradients are $\boldsymbol{G} \in \mathbb{R}^{n \times p}$. Then, we project the private gradients into the anchor subspace $\boldsymbol{B}$. The projection produces low-dimensional embeddings $\boldsymbol{W} = \boldsymbol{G}\boldsymbol{B}^T$ and residual gradients $\boldsymbol{R} = \boldsymbol{G} - \boldsymbol{G}\boldsymbol{B}^T\boldsymbol{B}$. The magnitude of residual gradients is usually much smaller than original gradient even when $k$ is small because of the gradient redundancy.

Then, we aggregate the gradient embeddings and the residual gradients, respectively. We perturb the aggregated embedding and the aggregated residual gradient respectively to guarantee certain differential privacy. Finally, we release the perturbed embedding and the perturbed residual gradient and construct an unbiased estimator of the private gradient: $\tilde{\boldsymbol{v}} := (\tilde{\boldsymbol{w}}^T\boldsymbol{B} + \tilde{\boldsymbol{r}})/n$. This construction process does not resulting in additional privacy loss because of DP's post-processing property. The privacy analysis of the whole process of GEP is given in Theorem 3.1.

**Theorem 3.1.** *Let $S_1$ and $S_2$ be the sensitivity of $\boldsymbol{w}$ and $\boldsymbol{r}$, respectively, the output of Algorithm 1 satisfies ($\epsilon, \delta$)-DP for any $\delta \in (0, 1)$ and $\epsilon \leq 2\log(1/\delta)$ if we choose $\sigma_1 \geq 2S_1\sqrt{2\log(1/\delta)}/\epsilon$ and $\sigma_2 \geq 2S_2\sqrt{2\log(1/\delta)}/\epsilon$.*

---

**Algorithm 1:** Gradient embedding perturbation

---

1: **Input:** anchor gradients $G^{(a)} \in \mathbb{R}^{m \times p}$; number of basis vectors $k$; private gradients $G \in \mathbb{R}^{n \times p}$; clipping thresholds $S_1, S_2$; standard deviations $\sigma_1, \sigma_2$; number of power iterations $t$.

2: *//First stage: Compute an orthonormal basis for the anchor subspace.*
3: Initialize $B \in \mathbb{R}^{k \times p}$ randomly.
4: **for** $i = 1$ **to** $t$ **do**
5:     Compute $A = G^{(a)} B^T$ and $B = A^T G^{(a)}$.
6:     Orthogonalize $B$ and normalize row vectors.
7: **end for**
8: Delete $G^{(a)}$ to free memory.

9: *//Second stage: project the private gradients $G$ into anchor subspace $B$*
10: Compute gradient embeddings $W = GB^T$ and clip its rows with $S_1$ to obtain $\hat{W}$.
11: Compute residual gradients $R = G - WB$ and clip its rows with $S_2$ to obtain $\hat{R}$.

12: *//Third stage: perturb gradient embedding and residual gradient separately*
13: Perturb embedding with noise $z^{(1)} \sim \mathcal{N}(0, \sigma_1^2 I_{k \times k})$: $w := \sum_i \hat{W}_{i,:}$, $\tilde{w} := w + z^{(1)}$.
14: Perturb residual gradient with noise $z^{(2)} \sim \mathcal{N}(0, \sigma_2^2 I_{p \times p})$: $r := \sum_i \hat{R}_{i,:}$, $\tilde{r} := r + z^{(2)}$.
15: Return $\tilde{v} := (\tilde{w}^T B + \tilde{r})/n$.

---

A common practice to control sensitivity is to clip the output with a pre-defined threshold. In our experiments, we use different thresholds $S_1$ and $S_2$ to clip the gradient embeddings and residual gradients, respectively. The privacy loss of GEP consists of two parts: the privacy loss incurred by releasing the perturbed embedding and the privacy loss incurred by releasing the perturbed residual gradient. We compose these two parts via the Rényi differential privacy and convert it to $(\epsilon, \delta)$-DP.

We highlight several implementation techniques that make GEP widely applicable and implementable with reasonable computational cost. Firstly, auxiliary non-sensitive data do not have to be the same source as the private data and the auxiliary data can be randomly labeled. This non-sensitive data assumption is very weak and easy to satisfy in practical scenarios. To understand why random label works, a quick example is that for the least squares regression problem the individual gradient is aligned with the feature vector while the label only scales the length but does not change the direction. This auxiliary data assumption avoids conducting principal component analysis (PCA) on private gradients, which requires releasing private high-dimensional basis vectors and hence introduces large privacy loss. Secondly, we use *power method* (Panju, 2011; Vogels et al., 2019) to approximately estimate the principal components. The new operation we introduce is standard matrix multiplication that enjoys efficient implementation on GPU. The computational complexity of each power iteration is $2mkp$, where $p$ is the number of model parameters, $m$ is the number of anchor gradients and $k$ is the number of subspace basis vectors. Thirdly, we divide the parameters into different groups and compute one orthonormal basis for each group. This further reduces the computational cost. For example, suppose the parameters are divided into two groups with size $p_1, p_2$ and the numbers of basis vectors are $k_1, k_2$, the computational complexity of each power iteration is $2m(k_1 p_1 + k_2 p_2)$, which is smaller than $2m(k_1 + k_2)(p_1 + p_2)$. In Appendix B, we analyze the additional computational and memory costs of GEP compared to standard gradient perturbation.

Curious readers may wonder if we can use random projection to reduce the dimensionality as Johnson–Lindenstrauss Lemma (Dasgupta & Gupta, 2003) guarantees that one can preserve the pairwise distance between any two points after projecting into a random subspace of much lower dimension. However, preserving the pairwise distance is not sufficient for high quality gradient reconstruction, which is verified by the empirical observation in Appendix C.

## 3.2 An analysis on the residual gradients of GEP

Let $g := \frac{1}{n} \sum_i G_{i,:}$ be the target private gradient. For a given anchor subspace $B$, the residual gradients are defined as $R := G - GB^T B$. We then analyze how large the residual gradients could be. The following argument holds for all time steps and we ignore the time step index for simplicity.

For the ease of discussion, we introduce $\boldsymbol{\xi}_i := (\boldsymbol{G}_{i,:})^T$ for $i \in [n]$ to denote the the private gradients and the $\hat{\boldsymbol{\xi}}_j := (\boldsymbol{G}_{j,:}^{(a)})^T$ for $j \in [m]$ to denote the anchor gradients. We use $\lambda_k(\cdot)$ to denote the $k_{th}$ largest eigenvalue of a given matrix. We assume that the private gradients $\boldsymbol{\xi}_1, ..., \boldsymbol{\xi}_n$ and the anchor gradients $\hat{\boldsymbol{\xi}}_1, ..., \hat{\boldsymbol{\xi}}_m$ are sampled independently from a distribution $\mathcal{P}$. We assume $\boldsymbol{\Sigma} := \mathbb{E}_{\boldsymbol{\xi} \sim \mathcal{P}} \boldsymbol{\xi} \boldsymbol{\xi}^T \in \mathbb{R}^{p \times p}$ to be the population gradient (uncentered) covariance matrix. We also consider the (uncentered) empirical gradient covariance matrix $\hat{\boldsymbol{S}} := \frac{1}{m} \sum_{i=1}^m \hat{\boldsymbol{\xi}}_i \hat{\boldsymbol{\xi}}_i^T$.

One case is that the population gradient covariance matrix $\boldsymbol{\Sigma}$ is low-rank $k$. In this case we can argue that the residual gradients are 0 once the number of anchor gradients $m > k$.

**Lemma 3.1.** *Assume that the population covariance matrix $\boldsymbol{\Sigma}$ is with rank $k$ and the distribution $\mathcal{P}$ satisfies $\mathbb{P}(\boldsymbol{\xi} \in \mathbb{F}_s) = 0$ for all $s$-flats $\mathbb{F}_s$ in $\mathbb{R}^p$ with $0 \leq s < k$. Let $\boldsymbol{\Sigma} = \boldsymbol{V}_k \Lambda \boldsymbol{V}_k^T$ and $\hat{\boldsymbol{S}} = \hat{\boldsymbol{V}}_{k'} \hat{\Lambda} \hat{\boldsymbol{V}}_{k'}^T$ be the eigendecompositions of $\boldsymbol{\Sigma}$ and the empirical covariance matrix $\hat{\boldsymbol{S}}$, respectively, such that $\lambda_{k'}(\hat{\boldsymbol{S}}) > 0$ and $\lambda_{k'+1}(\hat{\boldsymbol{S}}) = 0$. Then if $m \geq k$, we have with probability 1,*

$$k' = k \quad and \quad \|\boldsymbol{V}_k \boldsymbol{V}_k^T - \hat{\boldsymbol{V}}_k \hat{\boldsymbol{V}}_k^T\|_2 = 0. \tag{2}$$

*Proof.* The proof is based on the non-singularity of covariance matrix. See Appendix D. $\square$

We note that $s$-flat is the translate $\mathbb{F}_s = \boldsymbol{x} + \mathbb{F}_{s(0)}$ of an $s$-dimensional linear subspace $\mathbb{F}_{s(0)}$ in $\mathbb{R}^p$ and the normal distribution satisfies such condition (Eaton & Perlman, 1973; Muirhead, 2009). Therefore, we have seen that for low-rank case of population covariance matrix, the residual gradients are 0 once $m > k$. In the general case, we measure the expected norm of the residual gradients.

**Lemma 3.2.** *Assume that $\boldsymbol{\xi} \sim \mathcal{P}$ and $\|\boldsymbol{\xi}\|^2 < T$ almost surely. Let $\boldsymbol{\Sigma} = \boldsymbol{V} \Lambda \boldsymbol{V}^T$ be the eigendecomposition of the population covariance matrix $\boldsymbol{\Sigma}$. Let $\hat{\boldsymbol{S}} = \hat{\boldsymbol{V}}_k \hat{\Lambda} \hat{\boldsymbol{V}}_k^T$ be the eigendecomposition of the empirical covariance matrix $\hat{\boldsymbol{S}}$. Then we have with probability $1 - 2 \exp(-\delta)$,*

$$\mathbb{E} \|\boldsymbol{\xi} - \Pi_{\hat{\boldsymbol{V}}_k}(\boldsymbol{\xi})\|_2^2 \leq \sum_{k' > k} \lambda_{k'}(\boldsymbol{\Sigma}) + \sqrt{kC/m} + T\sqrt{2\delta/m}, \tag{3}$$

*where $C = \left[\mathbb{E}\|\boldsymbol{\xi}\|^4 - \sum_i \lambda_i^2(\boldsymbol{\Sigma})\right] + \left[\frac{1}{m} \sum_{j=1}^m \|\hat{\boldsymbol{\xi}}_j\|^4 - \sum_i \lambda_i^2(\hat{\boldsymbol{S}})\right]$, $\Pi_{\hat{\boldsymbol{V}}_k}$ is a projection operator onto the subspace $\hat{\boldsymbol{V}}_k$ and the $\mathbb{E}$ is taken over the randomness of $\boldsymbol{\xi} \sim \mathcal{P}$.*

*Proof.* The proof is an adaptation of Theorem 3.1 in Blanchard et al. (2007). $\square$

From Lemma 3.2, we can see the larger the number of anchor gradients and the dimension of the anchor subspace $k$, the smaller the residual gradients. We can choose $m, k$ properly such that the upper bound on the expected residual gradient norm is small. This indicates that we may use a smaller clipping threshold and consequently apply smaller noises with achieving the same privacy guarantee.

We next empirically examine the projection error $\boldsymbol{r} = \sum_i \boldsymbol{R}_{i,:}$ by training a 20-layer ResNet on CIFAR10 dataset. We try two different types of auxiliary data to compute the anchor gradients: 1) samples from the same source as private data with correct labels, i.e., 2000 random samples from the test data; 2) samples from different source with random labels, i.e., 2000 random samples from ImageNet. The relation between the dimension of anchor subspace $k$ and the projection error rate ($\left\| \frac{1}{n} \boldsymbol{r} \right\| / \|\boldsymbol{g}\|$) is presented in Figure 4. We can see that the project error is small and decreases with $k$, and the benefit of increasing $k$ diminishes when $k$ is large, which is implied by Lemma 3.2. In practice one can only use small or moderate $k$ because of the memory constraint. GEP needs to store at least $k$ individual gradients and each individual gradient consumes the same amount of memory as the model itself. Moreover, we can see that the projection into anchor subspace of random labeled auxiliary data yields comparable projection error, corroborating our argument that unlabeled auxiliary data are sufficient for finding the anchor subspace.

We also verify that the redundancy of residual gradients is small, by plotting the stable rank of residual gradient matrix in Figure 5. The stable rank of residual gradient matrix is an order of magnitude higher than the stable rank of original gradient matrix. This implies that it could be hard to further approximate $\boldsymbol{R}$ with low-dimensional embeddings.

We next compare the GEP with a scheme that simply discards the residual gradients and only outputs the perturbed gradient embedding, i.e., the output is $\tilde{\boldsymbol{u}} := \tilde{\boldsymbol{w}}^T \boldsymbol{B}/n$.

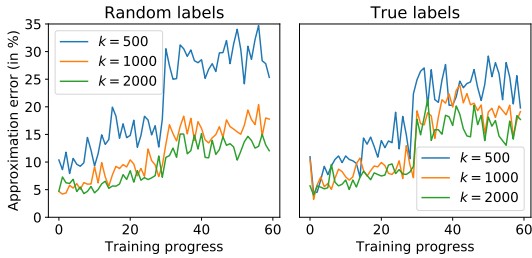 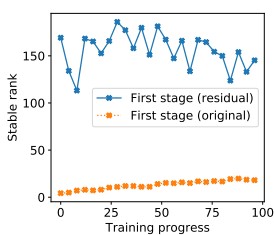

Figure 4: Relative projection error ($\left\| \frac{1}{n} \boldsymbol{r} \right\| / \|\boldsymbol{g}\|$) of the second stage in ResNet20. The number of anchor gradients is 2000. The dimension of anchor subspace is $k$. The learning rate is decayed by 10 at epoch 30. The left plot uses random samples from ImageNet. The right plot uses random samples from test data. The benefit of increasing $k$ becomes smaller when $k$ is larger.

Figure 5: Stable rank of the residual gradient matrix versus original gradient matrix. The gradients are computed on full batch data for the first stage in ResNet20. The dimension of anchor subspace is $k = 1000$.

**Remark 1.** *Let $\tilde{\boldsymbol{u}} := \tilde{\boldsymbol{w}}^T \boldsymbol{B}/n$ be the reconstructed gradient from noisy gradient embedding and $\tilde{\boldsymbol{v}}$ be the output of GEP. If ignoring the effect of gradient clipping, we have*

$$\mathbb{E}[\tilde{\boldsymbol{u}}] = \boldsymbol{g} - \boldsymbol{r}/n, \quad \mathbb{E}[\tilde{\boldsymbol{v}}] = \boldsymbol{g}. \tag{4}$$

*where $\boldsymbol{r} = \sum_i \boldsymbol{R}_{i,:}$ is the aggregated residual gradients, $\tilde{\boldsymbol{w}}, \boldsymbol{B}$ are given in Algorithm 1 and the expectation is over the added random noises.*

This indicates that $\tilde{\boldsymbol{u}}$ contains a systematic error that makes $\tilde{\boldsymbol{u}}$ always deviate from $\boldsymbol{g}$ by the residual gradient. This systematic error is the projection error, which is plotted in Figure 4. The systematic error cannot be mitigated by reducing the noise magnitude (e.g., increasing the privacy budget or collecting more private data). We refer to the algorithm releasing $\tilde{\boldsymbol{u}}$ directly as *Biased-GEP* or *B-GEP* for short, which can be viewed as an efficient implementation of the algorithm in (Zhou et al., 2020). In our experiments, B-GEP can outperform standard gradient perturbation when $k$ is large but is inferior to GEP. We note that the above remark is made with ignoring the clipping effect (or set a large clipping threshold). In practice, we do apply clipping for the individual gradients at each time step, which makes the expectations in Remark 1 obscure (Chen et al., 2020b). We note that the claim that $\tilde{\boldsymbol{v}}$ is an unbiased estimator of $\boldsymbol{g}$ is not that precise when applying gradient clipping.

### 3.3 PRIVATE LEARNING WITH GRADIENT EMBEDDING PERTURBATION

GEP (Algorithm 1) describes how to release one-step gradient with privacy guarantee. In this section, we compose the privacy losses at each step to establish the privacy guarantee for the whole learning process. The differentially private learning process with GEP is given in Algorithm 2 and the privacy analysis is presented in Theorem 3.2.

---

**Algorithm 2:** Differentially private gradient descent with GEP.

1: **Input:** private dataset $\mathbb{D}$; auxiliary dataset $\mathbb{D}^{(a)}$; number of updates $T$; learning rate $\eta$; configuration of GEP $\mathbb{C}$; loss function $\ell$;
2: **Output:** Differentially private model $\boldsymbol{\theta}_T$.
3: **for** $t = 0$ **to** $T - 1$ **do**
4:      Compute the private gradients $\boldsymbol{G}_t$ and anchor gradients $\boldsymbol{G}_t^{(a)}$ of loss with respect to $\boldsymbol{\theta}_t$.
5:      Call GEP with $\boldsymbol{G}_t, \boldsymbol{G}_t^{(a)}$ and configuration $\mathbb{C}$ to get $\tilde{\boldsymbol{v}}_t$.
6:      Update model $\boldsymbol{\theta}_{t+1} = \boldsymbol{\theta}_t - \eta \tilde{\boldsymbol{v}}_t$.
7: **end for**

---

**Theorem 3.2.** *For any $\epsilon < 2\log(1/\delta)$ and $\delta \in (0, 1)$, the output of Algorithm 2 satisfies $(\epsilon, \delta)$-DP if we set $\sigma \geq 2\sqrt{2T\log(1/\delta)}/\epsilon$.*

If the private gradients are randomly sampled from the full batch gradients, the privacy guarantee can be strengthened via the *privacy amplification by subsampling* theorem of DP (Balle et al., 2018; Wang et al., 2019; Zhu & Wang, 2019; Mironov et al., 2019). Theorem 3.3 gives the expected excess error of Algorithm 2. Expected excess error measures the distance between the algorithm's output and the optimal solution in expectation.

**Theorem 3.3.** *Suppose the loss $L(\boldsymbol{\theta}) = \frac{1}{n}\sum_{(\boldsymbol{x},y)\in\mathbb{D}}\ell(f_{\boldsymbol{\theta}}(\boldsymbol{x}), y)$ is 1-Lipschitz, convex, and $\beta$-smooth. If $\eta = \frac{1}{\beta}$, $T = \frac{n\beta\epsilon}{\sqrt{p}}$, and $\bar{\boldsymbol{\theta}} = \frac{1}{T}\sum_{t=1}^{T}\boldsymbol{\theta}_t$, then we have $\mathbb{E}[L(\bar{\boldsymbol{\theta}})] - L(\boldsymbol{\theta}_*) \leq \mathcal{O}\left(\frac{\sqrt{k\log(1/\delta)}}{n\epsilon} + \frac{\bar{r}\sqrt{p\log(1/\delta)}}{n\epsilon}\right)$, where $\bar{r} = \frac{1}{T}\sum_{t=0}^{T-1} r_t^2$ and $r_t = \max_i \|(\boldsymbol{R}_t)_{i,:}\|$ is the sensitivity of residual gradient at step $t$.*

The $\bar{r}$ term represents the average projection error over the training process. The previous best expected excess error for gradient perturbation is $\mathcal{O}(\sqrt{p\log(1/\delta)}/(n\epsilon))$ (Wang et al., 2017). As shown in Lemma 3.1, if the gradients locate in a $k$-dimensional subspace over the training process, $\bar{r} = 0$ and the excess error is $\mathcal{O}(\sqrt{k\log(1/\delta)}/(n\epsilon))$, independent of the problem ambient dimension $p$. When the gradients are in general position, i.e., gradient matrix is not exact low-rank, Lemma 3.2 and the empirical result give a hint on how small the residual gradients could be. However, it is hard to get a good bound on $\max_i \|(\boldsymbol{R}_t)_{i,:}\|$ and the bound in Theorem 3.3 does not explicitly improve over previous result. One possible solution is to use a clipping threshold based on the expected residual gradient norm. Then the output gradient becomes biased because of clipping and the utility/privacy guarantees in Theorem 3.3/3.2 require new elaborate derivation. We leave this for future work.

## 4 EXPERIMENTS

We conduct experiments on MNIST, extended SVHN, and CIFAR-10 datasets. Our implementation is publicly available[2]. The model for MNIST has two convolutional layers with max-pooling and one fully connected layer. The model for SVHN and CIFAR-10 is ResNet20 in He et al. (2016). We replace all batch normalization (Ioffe & Szegedy, 2015) layers with group normalization (Wu & He, 2018) layers because batch normalization mixes the representations of different samples and makes the privacy loss cannot be analyzed accurately. The non-private accuracy for MNIST, SVHN, and CIFAR-10 is 99.1%, 95.9%, and 90.4%, respectively.

We also provide experiments with pre-trained models in Appendix A. Tramèr & Boneh (2020) show that differentially private linear classifier can achieve high accuracy using the features produced by pre-trained models. We examine whether GEP can improve the performance of such private linear classifiers. Notably, using the features produced by a model pre-trained on unlabeled ImageNet, GEP achieves 94.8% validation accuracy on CIFAR10 with $\epsilon = 2$.

**Evaluated algorithms** We use the algorithm in Abadi et al. (2016) as benchmark gradient perturbation approach, referred to as "GP". We also compare GEP with PATE (Papernot et al., 2017). We run the experiments for PATE using the official implementation. The privacy parameter $\epsilon$ of PATE is data-dependent and hence cannot be released directly (see Section 3.3 in Papernot et al. (2017)). Nonetheless, we report the results of PATE anyway.

**Implementation details** At each step, GEP needs to release two vectors: the noisy gradient embedding and the noisy residual gradient. The gradient embeddings have a sensitivity of $S_1$ and the residual gradients have a sensitivity of $S_2$ because of the clipping. The output of GEP can be constructed as follows: (1) normalize the gradient embeddings and residual gradients by $1/S_1$ and $1/S_2$, respectively, (2) concatenate the rescaled vectors, (3) release the concatenated vector via gaussian mechanism with sensitivity $\sqrt{2}$, (4) rescale the two components by $S_1$ and $S_2$. B-GEP only needs to release the normalized noisy gradient embedding. We use the numerical tool in Mironov et al. (2019) to compute the privacy loss. For given privacy budget and sampling probability, $\sigma$ is set to be the smallest value such that the privacy budget is allowable to run desired epochs.

All experiments are run on a single Tesla V100 GPU with 16G memory. For ResNet20, the parameters are divided into five groups: input layer, output layer, and three intermediate stages. For a given quota of basis vectors, we allocate it to each group according to the square root of the number of parameters in each group. We compute an orthonormal subspace basis on each group separately.

---

[2]https://github.com/dayu11/Gradient-Embedding-Perturbation

Table 1: Test accuracy (in %) with varying choices of privacy bound $\epsilon$. The numbers under symbol $\Delta$ denote the improvement over GP baseline.

| Dataset | Algorithm | $\epsilon = 2$ | $\Delta$ | $\epsilon = 5$ | $\Delta$ | $\epsilon = 8$ | $\Delta$ |
|---------|-----------|------|------|------|------|------|------|
| MNIST | GP | 94.7 | +0.0 | 96.8 | +0.0 | 97.2 | +0.0 |
| | PATE | 98.5 | **+3.8** | 98.5 | **+1.7** | 98.6 | **+1.4** |
| | B-GEP | 93.1 | -1.6 | 94.5 | -2.3 | 95.9 | -1.3 |
| | GEP | 96.3 | +1.6 | 97.9 | **+1.1** | 98.4 | **+1.2** |
| SVHN | GP | 87.1 | +0.0 | 91.3 | +0.0 | 91.6 | +0.0 |
| | PATE | 80.7 | -6.4 | 91.6 | +0.3 | 91.6 | +0.0 |
| | B-GEP | 88.5 | +1.4 | 91.8 | +0.5 | 92.3 | +0.7 |
| | GEP | 92.3 | **+5.2** | 94.7 | **+3.4** | 95.1 | **+3.5** |
| CIFAR-10 | GP | 43.6 | +0.0 | 52.2 | +0.0 | 56.4 | +0.0 |
| | PATE | 34.2 | -9.4 | 41.9 | -10.3 | 43.6 | -12.8 |
| | B-GEP | 50.3 | +6.7 | 59.5 | +7.3 | 63.0 | +6.6 |
| | GEP | 59.7 | **+16.1** | 70.1 | **+17.9** | 74.9 | **+18.5** |

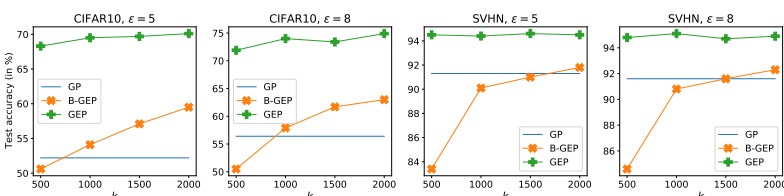

Figure 6: Test accuracy when varying the dimension of anchor subspace. GEP significantly outperforms B-GEP for all $k$. Moreover, the performance of GEP is not that sensitive to $k$.

Then we concatenate the projections of all groups to construct gradient embeddings. The number of power iterations $t$ is set as 1 as empirical evaluations suggest more iterations do not improve the performance for GEP and B-GEP.

For all datasets, the anchor gradients are computed on 2000 random samples from ImageNet. In Appendix C, we examine the influence of choosing different numbers of anchor gradients and different sources of auxiliary data. The selected images are downsampled into size of $32 \times 32$ ($28 \times 28$ for MNIST) and we label them randomly at each update. For SVHN and CIFAR-10, $k$ is chosen from $[500, 1000, 1500, 2000]$. For MNIST, we halve the size of $k$. We use SGD with momentum 0.9 as the optimizer. Initial learning rate and batchsize are 0.1 and 1000, respectively. The learning rate is divided by 10 at middle of training. Weight decay is set as $1 \times 10^{-4}$. The clipping threshold for is 10 for original gradients and 2 for residual gradients. The number of training epochs for CIFAR-10 and MNIST is 50, 100, 200 for privacy parameter $\epsilon = 2, 5, 8$, respectively. The number of training epochs for SVHN is 5, 10, 20 for privacy parameter $\epsilon = 2, 5, 8$, respectively. Privacy parameter $\delta$ is $1 \times 10^{-6}$ for SVHN and $1 \times 10^{-5}$ for CIFAR-10 and MNIST.

**Results** The best accuracy with given $\epsilon$ is in Table 4. For all datasets, GEP achieves considerable improvement over GP in Abadi et al. (2016). Specifically, GEP achieves $74.9\%$ test accuracy on CIFAR-10 with $(8, 10^{-5})$-DP, outperforming GP by $18.5\%$. PATE achieves best accuracy on MNIST but its performance drops as the dataset becomes more complex.

We also plot the relation between accuracy and $k$ in Figure 6. GEP is less sensitive to the choice of $k$ and outperforms B-GEP for all choices of $k$. The improvement of increasing $k$ becomes smaller as $k$ becomes larger. We note that the memory cost of choosing large $k$ is high because we need to store at least $k$ individual gradients to compute anchor subspace.

## 5 CONCLUSION

In this paper, we propose Gradient Embedding Perturbation (GEP) for learning with differential privacy. GEP leverages the gradient redundancy to reduce the added noise and outputs an unbiased estimator of target gradient. The several key designs of GEP significantly boost the applicability of GEP. Extensive experiments on real world datasets demonstrate the superior utility of GEP.

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

## A    EXPERIMENTS WITH PRE-TRAINED MODELS

Recent works have shown that pre-training the models on unlabeled data can be beneficial for subsequent learning tasks (Chen et al., 2020a; He et al., 2020). Tramèr & Boneh (2020) demonstrate that differentially private linear classifier can achieve high accuracy using the features produced by those per-trained models. We show that GEP can also benefit from such pre-trained models.

Inspired by Tramèr & Boneh (2020), we use the output of the penultimate layer of a pre-trained ResNet152 model as feature to train a private linear classifier. The ResNet152 model is pre-trained on unlabeled ImageNet using SimCLR (Chen et al., 2020a). The feature dimension is 4096.

**Implementation Details** We choose the privacy parameter $\epsilon$ from $[0.1, 0.5, 1, 2]$. The privacy parameter $\delta$ is $1 \times 10^{-5}$. We run all experiments for 5 times and report the average accuracy. The clipping threshold of residual gradients is still one-fifth of the clipping threshold of the original gradients. The dimension of anchor subspace is set as $200 \simeq \sqrt{p}$ where $p = 40960$ is the model dimension. We randomly sample 500 samples from the test set as auxiliary data and evaluate performance on the rest test samples. The optimizer is Adam with default momentum coefficients. Other hyper-parameters are listed in Table 2.

| Hyperparameter | Values |
|---|---|
| Learning rate | 0.01, 0.05, 0.1 |
| Running steps | 50, 100, 400 |
| Clipping threshold | 0.01, 0.1, 1 |

Table 2: Hyperparameter values used in Appendix A.

**Results** The experiment results are shown in Table 3. GEP outperforms GP on all values of $\epsilon$. With privacy bound $\epsilon = 2$, GEP achieves 94.8% validation accuracy on CIFAR10 dataset, improving over the GP baseline by 1.4%. For very strong privacy guarantee ($\epsilon = 0.1$), B-GEP performs on par with GEP because strong privacy guarantee requires large noise and the useful signal in residual gradient is submerged in the added noise. B-GEP benefits less from larger $\epsilon$ compared to GP or GEP. For $\epsilon = 1$ and 2, the performance of B-GEP is worse than the performance of GP. This is because larger $\epsilon$ can not reduce the systematic error of B-GEP (see Remark 1 in Section 3.2).

Table 3: Validation accuracy (in %) on CIFAR10 with varying choices of $\epsilon$. We train a private linear model on top of the features from a ResNet152 model, which is pre-trained on unlabeled ImageNet.

| | $\epsilon = 0.1$ | $\epsilon = 0.5$ | $\epsilon = 1$ | $\epsilon = 2$ |
|---|---|---|---|---|
| Non private | 96.3 | 96.3 | 96.3 | 96.3 |
| GP | 88.2 ($\pm$0.16) | 91.1 ($\pm$0.17) | 93.2 ($\pm$0.19) | 93.4 ($\pm$0.12) |
| B-GEP | **91.0** ($\pm$0.07) | 92.9 ($\pm$0.03) | 93.1 ($\pm$0.10) | 93.2 ($\pm$0.08) |
| GEP | 90.9 ($\pm$0.19) | **93.5** ($\pm$0.06) | **94.3** ($\pm$0.09) | **94.8** ($\pm$0.06) |

## B    COMPLEXITY ANALYSIS

We provide an analysis of the computational and memory costs of the construction of anchor subspace. The computation of the anchor subspace is the dominant additional cost of GEP compared to conventional gradient perturbation. Notations: $k$, $m$, $n$, and $p$ are the dimension of anchor subspace, number of anchor gradients, number of private gradients, and the model dimension, respectively. In order to reduce the computational and memory costs, we divide the parameters into $g$ groups and compute one orthonormal basis for each group. We refer to this approach as 'parameter grouping'. In this section, we assume the parameters and the dimension of the anchor subspace are both divided evenly. Table 4 summarizes the additional costs of GEP with/without parameter grouping. Using parameter grouping can reduce the computational/memory cost significantly.

Table 4: Computational and memory costs of a single power iteration in Algorithm 1. The computation cost is measured by the number of floating point operations. The memory cost is measured by the number of floating-point numbers we need to store. 'GEP+PG' denotes GEP with parameter grouping and $g$ denotes the number of groups. Notations: $k$, $m$, $n$, and $p$ are the dimension of anchor subspace, number of anchor gradients, number of private gradients, and the model dimension, respectively.

|  | Computational Cost | Memory Cost |
|---|---|---|
| GEP | $2mkp + pk^2$ | $\max\left(0, (m - n + k)\, p + mk\right)$ |
| GEP+PG | $2mkp/g + pk^2/g^2$ | $\max\left(0, \left(m - n + \frac{k}{g}\right) p + mk\right)$ |

## C  ABLATION STUDY

**The influence of choosing different auxiliary datasets.** We conduct experiments with different choices of auxiliary datasets. For CIFAR10, we try 2000 random test samples from CIFAR10, 2000 random samples from CIFAR100, and 2000 random samples from ImageNet. When the auxiliary dataset is CIFAR10, we try both correct labels and random labels. For all choices of auxiliary datasets, the test accuracy is evaluated on 8000 test samples of CIFAR10 that are not used as auxiliary data. Other implementation details are the same as in Section 4. The results are shown in Table 5. Surprisingly, using samples from CIFAR10 with correct labels yields the worst accuracy. This may because the model 'overfits' the auxiliary data when it has access to correct labels, which makes the anchor subspace contains less information about the private gradients. The best accuracy is achieved using samples from CIFAR10 with random labels, this makes sense because in this case the features of auxiliary data and private data have the same distribution. Using samples from CIFAR100 or ImageNet as auxiliary data has a small influence on the test accuracy.

Table 5: Test accuracy on CIFAR10 with different choices of auxiliary datasets. The privacy guarantee is $(8, 10^{-5})$-DP. We report the average accuracy of five runs with standard deviations in brackets.

| Auxiliary Data | Random Label? | Test Accuracy |
|---|---|---|
| CIFAR10 | No | 72.9 ($\pm$0.31) |
| CIFAR10 | Yes | **75.1** ($\pm$0.42) |
| CIFAR100 | Yes | 74.7 ($\pm$0.46) |
| ImageNet | Yes | 74.8 ($\pm$0.39) |

**The influence of the number of anchor gradients.** In the main text, the size of auxiliary dataset is $m = 2000$. We conduct more experiments with different sizes of auxiliary dataset to examine the influence of $m$. The auxiliary data is randomly sampled from ImageNet. Table 6 reports the test accuracy on CIFAR10 with different choices of $m$. For both B-GEP and GEP, increasing $m$ leads to slightly improved performance.

Table 6: Test accuracy on CIFAR10 with different sizes of auxiliary dataset. The privacy guarantee is $(8, 10^{-5})$-DP. We report the average accuracy of five runs with standard deviations in brackets.

| Algorithm | $m = 1000$ | $m = 2000$ | $m = 4000$ |
|---|---|---|---|
| B-GEP | 62.2 ($\pm$0.26) | 62.6 ($\pm$0.24) | 63.3 ($\pm$0.27) |
| GEP | 74.6 ($\pm$0.41) | 74.8 ($\pm$0.39) | **75.2** ($\pm$0.34) |

**The projection error of random basis vectors.** It is tempting to construct the anchor subspace using random basis vectors because Johnson–Lindenstrauss Lemma (Dasgupta & Gupta, 2003) guarantees that one can preserve the pairwise distance between any two points after projecting into a random subspace of much lower dimension. We empirically verify the projection error of Gaussian random basis vectors on CIFAR10 and SVHN. The experiment settings are the same as in Section 4. The projection errors over the training process are plotted in Figure 7. The projection error of random basis vectors is very high ($> 95\%$) throughout training. This is because preserving the pairwise

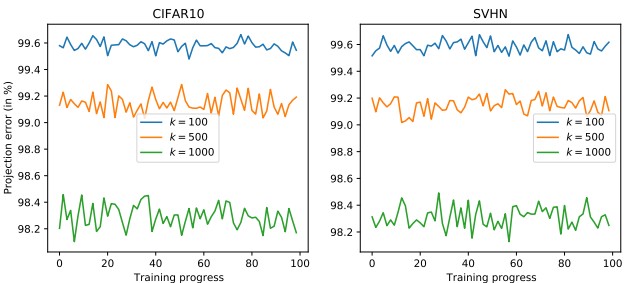

Figure 7: Projection error rate of random basis vectors. The dimension of subspace is denoted by $k$.

distance is not sufficient for high quality gradient reconstruction, which requires one to preserve the average 'distance' between any individual gradient and all other gradients.

# D    MISSING PROOFS

**Lemma 3.1.** *Assume that the population covariance matrix $\Sigma$ is with rank $k$ and the distribution $\mathcal{P}$ satisfies $\mathbb{P}(\xi \in \mathbb{F}_s) = 0$ for all $s$-flats $\mathbb{F}_s$ in $\mathbb{R}^p$ with $0 \leq s < k$. Let $\Sigma = V_k \Lambda V_k^T$ and $\hat{S} = \hat{V}_{k'} \hat{\Lambda} \hat{V}_{k'}^T$ be the eigendecompositions of $\Sigma$ and the empirical covariance matrix $\hat{S}$, respectively, such that $\lambda_{k'}(\hat{S}) > 0$ and $\lambda_{k'+1}(\hat{S}) = 0$. Then if $m \geq k$, we have with probability 1,*

$$k' = k \quad and \quad \|V_k V_k^T - \hat{V}_k \hat{V}_k^T\|_2 = 0. \tag{2}$$

*Proof.* We extend the Theorem 3.2 in Eaton & Perlman (1973) to the low-rank case.

**Theorem D.1** (Theorem 3.2 in Eaton & Perlman (1973)). *Let $X = (x_1, ..., x_n)$ where the $x_i$ are i.i.d. random vectors in $\mathbb{R}^p$, $n \geq p$. If $\mathbb{P}\{x_1 \in \mathbb{M}\} = 0$ for all proper manifolds $\mathbb{M} \subset \mathbb{R}^p$, then $\mathbb{P}\{X$ is non-singular$\}$=1.*

We note that the subspace spanned by $\hat{V}_{k'}$ is in the space spanned by $V_k$ by definition. Hence $k' \leq k$.

Let $\hat{x}_i := V_k^T \hat{\xi}_i \in \mathbb{R}^k$ for $i \in [m]$. Then $\hat{X} := (\hat{x}_1, ..., \hat{x}_m)$ is non-singular because of the assumption and Theorem D.1. That is $rank(\hat{X}) = k$. Therefore $rank((\hat{\xi}_1, ..., \hat{\xi}_m)) \geq k$, $rank(\hat{S}) \geq k$ and $k' \geq k$. Therefore $k' = k$ and the subspace spanned by $\hat{V}_{k'}$ and the subspace spanned by $V_k$ are identical. □

**Theorem 3.1.** *Let $S_1$ and $S_2$ be the sensitivity of $w$ and $r$, respectively, the output of Algorithm 1 satisfies $(\epsilon, \delta)$-DP for any $\delta \in (0, 1)$ and $\epsilon \leq 2\log(1/\delta)$ if we choose $\sigma_1 \geq 2S_1\sqrt{2\log(1/\delta)}/\epsilon$ and $\sigma_2 \geq 2S_2\sqrt{2\log(1/\delta)}/\epsilon$.*

*Proof of Theorem 3.1.* We first introduce some background knowledge of Rényi differential privacy (RDP) (Mironov, 2017). RDP measures the Rényi divergence between two output distributions.

**Definition 2** $((\lambda, \gamma)$-RDP). *A randomized mechanism $f$ is said to guarantee $(\lambda, \gamma)$-RDP if for any neighboring datasets $\mathbb{D}, \mathbb{D}'$ and $\lambda > 1$ it holds that*

$$D_\lambda(f(\mathbb{D})\|f(\mathbb{D}')) \leq \gamma,$$

*where $D_\lambda(\cdot\|\cdot)$ denotes the Rényi divergence of order $\lambda$.*

We next introduce some useful properties of RDP.

**Lemma D.2** (Gaussian mechanism of RDP). *Let $S = \max_{\mathbb{D} \sim \mathbb{D}'} \|f(\mathbb{D}) - f(\mathbb{D}')\|$ be the $l_2$ sensitivity, then Gaussian mechanism $\mathcal{M} = f(\mathbb{D}) + z$ satisfies $(\lambda, \frac{\lambda S^2}{2\sigma^2})$-RDP, where $z \sim \mathcal{N}(0, \sigma^2 I_{p \times p})$.*

**Lemma D.3** (Composition of RDP). *If $M_1$, $M_2$ satisfy $(\lambda, \gamma_1)$-RDP and $(\lambda, \gamma_2)$-RDP respectively, then their composition satisfies $(\lambda, \gamma_1 + \gamma_2)$-RDP.*

**Lemma D.4** (Conversion from RDP to $(\epsilon, \delta)$-DP). *If $\mathcal{M}$ obeys $(\lambda, \gamma)$-RDP, then $\mathcal{M}$ obeys $(\gamma + \log(1/\delta)/(\lambda - 1), \delta)$-DP for all $0 < \delta < 1$.*

Now we proof Theorem 3.1. Let $\boldsymbol{W}, \boldsymbol{W}'$ be the gradient embeddings of two neighboring datasets $\mathbb{D} \sim \mathbb{D}'$ and $\boldsymbol{R}, \boldsymbol{R}'$ be corresponding residual gradients. Without loss of generality, suppose $\boldsymbol{W}$ ($\boldsymbol{R}$) has one more row than $\boldsymbol{W}'$ ($\boldsymbol{R}'$). For given sensitivity $S_1, S_2$,

$$\max_{\mathbb{D} \sim \mathbb{D}'} \|\boldsymbol{w} - \boldsymbol{w}'\| = \max_{\boldsymbol{W} \sim \boldsymbol{W}'} \|\boldsymbol{W}_{n,:}\| \leq S_1, \quad \max_{\mathbb{D} \sim \mathbb{D}'} \|\boldsymbol{r} - \boldsymbol{r}'\| = \max_{\boldsymbol{R} \sim \boldsymbol{R}'} \|\boldsymbol{R}_{n,:}\| \leq S_2.$$

If we set $\sigma_1 = S_1 \sigma$ and $\sigma_2 = S_2 \sigma$ for some $\sigma$, then Algorithm 1 satisfies $(\lambda, \frac{\lambda}{\sigma^2})$-RDP because of Lemma D.2 and D.3. In order to guarantee $(\epsilon, \delta)$-DP, we need

$$\frac{\lambda}{\sigma^2} + \frac{\log(1/\delta)}{\lambda - 1} \leq \epsilon. \tag{5}$$

Choose $\lambda = 1 + \frac{2 \log(1/\delta)}{\epsilon}$ and rearrange Eq (5), we need

$$\sigma^2 \geq \frac{2 \left(\epsilon + 2 \log(1/\delta)\right)}{\epsilon^2}. \tag{6}$$

Then using the constraint on $\epsilon$ concludes the proof.

$\square$

**Theorem 3.2.** *For any $\epsilon < 2 \log(1/\delta)$ and $\delta \in (0, 1)$, the output of Algorithm 2 satisfies $(\epsilon, \delta)$-DP if we set $\sigma \geq 2\sqrt{2T \log(1/\delta)}/\epsilon$.*

*Proof of Theorem 3.2.* From the proof of Theorem 3.1, we have each call of GEP satisfies $(\lambda, \frac{\lambda}{\sigma^2})$-RDP. Then by the composition property of RDP (Lemma D.3), the output of Algorithm 2 satisfies $(\lambda, \frac{T\lambda}{\sigma^2})$-RDP. Plugging $\frac{T\lambda}{\sigma^2}$ into Equation 5 and 6 concludes the proof.

$\square$

**Theorem 3.3.** *Suppose the loss $L(\boldsymbol{\theta}) = \frac{1}{n} \sum_{(\boldsymbol{x}, y) \in \mathbb{D}} \ell(f_{\boldsymbol{\theta}}(\boldsymbol{x}), y)$ is 1-Lipschitz, convex, and $\beta$-smooth. If $\eta = \frac{1}{\beta}$, $T = \frac{n\beta\epsilon}{\sqrt{p}}$, and $\bar{\boldsymbol{\theta}} = \frac{1}{T} \sum_{t=1}^{T} \boldsymbol{\theta}_t$, then we have $\mathbb{E}[L(\bar{\boldsymbol{\theta}})] - L(\boldsymbol{\theta}_*) \leq \mathcal{O}\left(\frac{\sqrt{k \log(1/\delta)}}{n\epsilon} + \frac{\bar{r}\sqrt{p \log(1/\delta)}}{n\epsilon}\right)$, where $\bar{r} = \frac{1}{T} \sum_{t=0}^{T-1} r_t^2$ and $r_t = \max_i \|(\boldsymbol{R}_t)_{i,:}\|$ is the sensitivity of residual gradient at step $t$.*

*Proof of Theorem 3.3.* The $\beta$-smooth condition gives

$$L(\boldsymbol{\theta}_{t+1}) \leq L(\boldsymbol{\theta}_t) + \langle \nabla L(\boldsymbol{\theta}_t), \boldsymbol{\theta}_{t+1} - \boldsymbol{\theta}_t \rangle + \frac{\beta}{2} \|\boldsymbol{\theta}_{t+1} - \boldsymbol{\theta}_t\|^2. \tag{7}$$

Based on the update rule of GEP we have

$$\boldsymbol{\theta}_{t+1} - \boldsymbol{\theta}_t = -\eta \tilde{\boldsymbol{v}} = -\eta \nabla L(\boldsymbol{\theta}_t) - \frac{\eta}{n}(\boldsymbol{z}_t^{(1)} \boldsymbol{B} + \boldsymbol{z}_t^{(2)}), \tag{8}$$

where $\boldsymbol{z}_t^{(1)} \sim \mathcal{N}(0, \sigma^2 \boldsymbol{I}_{k \times k})$, $\boldsymbol{z}_t^{(2)} \sim \mathcal{N}(0, \sigma^2 r_t^2 \boldsymbol{I}_{p \times p})$ are the perturbation noises and $r_t = \max_i \|(\boldsymbol{R}_t)_{i,:}\|$ is the sensitivity of residual gradients at step $t$.

Take expectation on Eq (7) with respect to the perturbation noises.

$$\mathbb{E}[L(\boldsymbol{\theta}_{t+1})] \leq \mathbb{E}[L(\boldsymbol{\theta}_t)] - (\eta - \beta\eta^2/2)\mathbb{E}[\|\nabla L(\boldsymbol{\theta}_t)\|^2] + \frac{\beta\eta^2\sigma^2}{2n^2}\left(k + pr_t^2\right). \tag{9}$$

Subtract $L(\boldsymbol{\theta}_*)$ from both sides, we have

$$\mathbb{E}[L(\boldsymbol{\theta}_{t+1})] - L(\boldsymbol{\theta}_*) \leq \mathbb{E}[L(\boldsymbol{\theta}_t)] - L(\boldsymbol{\theta}_*) - (\eta - \beta\eta^2/2)\mathbb{E}[\|\nabla L(\boldsymbol{\theta}_t)\|^2] + \frac{\beta\eta^2\sigma^2}{2n^2}\left(k + pr_t^2\right)$$

$$\leq \mathbb{E}[\langle \nabla L(\boldsymbol{\theta}_t), \boldsymbol{\theta}_t - \boldsymbol{\theta}_* \rangle] - (\eta - \beta\eta^2/2)\mathbb{E}[\|\nabla L(\boldsymbol{\theta}_t)\|^2] + \frac{\beta\eta^2\sigma^2}{2n^2}\left(k + pr_t^2\right). \tag{10}$$

The second inequality holds because $L$ is convex. Then choose $\eta = \frac{1}{\beta}$ and plug $\nabla L(\boldsymbol{\theta}_t) = (\boldsymbol{\theta}_t - \boldsymbol{\theta}_{t+1})/\eta - (\boldsymbol{z}_1^t \boldsymbol{B} + \boldsymbol{z}_2^t)/n$ into Eq (10).

$$
\begin{aligned}
\mathbb{E}[L(\boldsymbol{\theta}_{t+1})] - L(\boldsymbol{\theta}_*) &\leq \beta \mathbb{E}[\langle \boldsymbol{\theta}_t - \boldsymbol{\theta}_{t+1}, \boldsymbol{\theta}_t - \boldsymbol{\theta}_* \rangle] - \frac{\beta}{2} \mathbb{E}[\|\boldsymbol{\theta}_t - \boldsymbol{\theta}_{t+1}\|^2] + \frac{\sigma^2}{\beta n^2} \left( k + p r_t^2 \right) \\
&= \frac{\beta}{2} \left( \mathbb{E}[\|\boldsymbol{\theta}_t - \boldsymbol{\theta}_*\|^2] - \mathbb{E}[\|\boldsymbol{\theta}_{t+1} - \boldsymbol{\theta}_*\|^2] \right) + \frac{\sigma^2}{\beta n^2} \left( k + p r_t^2 \right).
\end{aligned}
\tag{11}
$$

Sum over $t = 0, \ldots, T - 1$ and use convexity, we have

$$
\mathbb{E}[L(\bar{\boldsymbol{\theta}})] - L(\boldsymbol{\theta}_*) \leq \frac{\beta}{2T} \|\boldsymbol{\theta}_0 - \boldsymbol{\theta}_*\| + \frac{\sigma^2}{\beta n^2}(k + \frac{p}{T} \sum_{t=0}^{T-1} r_t^2).
\tag{12}
$$

Then substituting $T = \frac{n \beta \epsilon}{\sqrt{p}}$ and $\sigma = \mathcal{O}(\sqrt{T \log(1/\delta)}/\epsilon)$ yields the desired bound.

$\square$

