# OpenReview forum: "Do not Let Privacy Overbill Utility:  Gradient Embedding Perturbation for Private Learning"
_ICLR.cc/2021/Conference — ICLR 2021 Poster_

### Official Review · AnonReviewer3 · 2020-10-25

**Rating:** 5
**Confidence:** 5

**Review:**

This paper studied how to use the information that the gradient lies in a low dimension space to design more accurate differentially private SGD algorithm. Specifically, the authors proposed a new method which is called Gradient Embedding perturbation method. And they showed that compared with the previous DP-SDG method, their method has higher accuracy.
Actually, I tend to weakly reject the paper due to the following reasons.
1. As I know and also the authors mentioned in the related work part, there are also two other works study how to use the low dimensional gradient to get higher utilities for the Empirical Risk Minimization problem. However, I can see that in the other two works, they all have theoretical results for the utility, while in this paper there is just a variance analysis, so I wish to see the authors use Theorem 3.2 to get the excess error for their method when the loss functions are convex so that we can see the difference between the other two methods clearly.
2. In the experimental part, I think it will be better to have a comparison with the above two papers to see the superiority of the method in this paper. just comparing with the base-line is not enough.  Moreover, the authors set $\delta=10^{-5}$ or $10^{-6}$, I think it will be better to set $\delta=\frac{1}{n}$. And I want to see more comments about the number of public data to get the anchor subspace.

---

> ### Author Response · Authors · 2020-11-17
> **Response to Reviewer 3 - Part 1**
>
> Thank you for your review and valuable comments.
>
> 1）Q: There is no theoretical result for the utility of the proposed method.
>
> A: In the updated submission, we give the expected excess error bound (Theorem 3.3 in Section 3.3) for learning with GEP in the standard convex setting. There are two terms on the right hand side: one is $\frac{k}{n\epsilon}$, where $k$ is the dimension of anchor subspace and the other is the previous best utility bound multiplied by a term representing “the average projection error over the training process”.   When the gradient matrices stay in the same low-rank subspace over the training procedure, the utility bound does not depend on the ambient dimension but only depends on $k$ by properly choosing $m$ and $k$.  In general case, our bound justifies the superiority of GEP because of Lemma 3.2 and empirical observation. We note that our bound is at least as good as the previous best bound under the same assumptions.
>
> We clarify the differences between GEP and the other two concurrent works.  [1] uses the gradients of some public samples to construct a low-dimensional subspace and then project the noisy private gradients into such subspace. Under some assumptions, e.g., population gradient covariance matrix is low-rank at each step (they do not require the low-rank subspace to stay the same during training), they give a utility bound that depends on the model dimension very mildly ($\log p$). Although our submission shares similar insight with [1] at a high level, we highlight several reasons that make our submission irreplaceable and valuable to the community.
> (1): The method in [1] (PDP-SGD) discards the residual component after projection, which makes their update contains a systematic error. **Such systematic error cannot be mitigated by reducing the noise variance (e.g., increasing the privacy budget or collecting more private data).**  This phenomenon is verified by the experiments in [1].
> (2): We introduce several techniques, e.g., using the power method for construction of the projection subspace, to improve the efficiency of the computation of the subspace. With the proposed techniques, the dimension of the subspace we can leverage is up to 2000 on a Resnet20 model when using a single GPU card with 16G memory. On a small convolutional network and GPU cards with 12G memory, the largest subspace dimension explored by [1] is only 50.
> (3): [1] assumes the public samples are from the same source as the private dataset while we show the auxiliary data does not need to be the same source as the private data, which makes the auxiliary data more accessible in realistic settings.
>
> In [2], the authors leverage the historical noisy gradients instead of public gradients to improve the utility bound of private AdaGrad. They show the utility bound is independent of the model dimension if the gradients stay in the same low-rank subspace throughout the training. GEP uses some auxiliary gradients instead of historical gradients to build the projection subspace. An interesting future direction is to study how to leverage both public gradients and historical gradients to compute the projection subspace.
>
> 2）Q：Empirical comparisons with two concurrent papers are needed.
>
> A：As for the PDP-SGD algorithm in [1], the B-GEP in our paper can serve as a substitute as both B-GEP and PDP-SGD discard the residual gradients. Differences between B-GEP and PDP-SGD  are the implementation and the construction of the projection subspace. B-GEP uses power methods to find the embedding subspace while PDP-SGD uses SVD. B-GEP splits the parameters into groups to reduce the memory cost. These techniques make B-GEP can accommodate a much larger subspace dimension than PDP-SGD. The benefit of larger subspace dimension is verified by [1] and Figure 6 in our paper. Therefore, B-GEP can be viewed as a stronger baseline than PDP-SGD.
> We summarize the comparison between B-GEP and GEP. GEP outperforms B-GEP in most of the experiments. When the privacy budget is very small (epsilon=0.1), B-GEP  is on par with GEP on the CIFAR10 dataset (see Appendix A). The extensive experiments in our paper justify the distinctive value of GEP compared to PDP-SGD.
> We do not include the comparison between GEP and the method in [2] in our paper because [2] requires the storage and inverse of a large matrix ($p\times p$, where $p$ is the model size), which is not possible to do for deep network objective. Moreover, [2] does not provide any details about the algorithm implementation nor open-source code.

---

> > ### Author Response · Authors · 2020-11-17
> > **Response to Reviewer 3 - Part 2**
> >
> > 3）Q：Privacy parameter $\delta$ should be set as $\frac{1}{n}$.
> >
> > A：The value of $\delta$ is set to follow previous works [3,4] for a fair comparison. Actually, the choices of $\delta$ in [3,4] are very close to $\frac{1}{n}$. For MNIST, CIFAR10, and SVHN, the values of $\frac{1}{n}$ are $1.7\times 10^{-5}$, $2\times 10^{-5}$, and $1.6\times 10^{-6}$, respectively. For the CIFAR10 classification task with $\epsilon=8$, using $\delta=2\times 10^{-5}$ only changes the noise scale from 2.09 to 2.04.
> >
> > 4）Q：More comments about the number of public data to get the anchor subspace.
> >
> > A：We conduct experiments with different numbers of public data to examine their influence. The experiment results are presented in Table 6 in Appendix C of the revised version. Increasing the size of public data $m$ leads to slightly improved performance.  The test accuracy of GEP on CIFAR10 with $m$=1000, 2000, and 4000 are 74.6%, 74.8%, and 75.2%, respectively. Increasing $m$ also increases the memory cost because one needs to store more anchor gradients. Hence, the choice of $m$ provides a trade-off between efficiency and accuracy in practice.
> >
> > [1]: Bypassing the Ambient Dimension: Private SGD with Gradient Subspace Identification
> >
> > [2]: Dimension Independence in Unconstrained Private ERM via Adaptive Preconditioning
> >
> > [3]: Deep Learning with Differential Privacy
> >
> > [4]: Semi-supervised Knowledge Transfer for Deep Learning from Private Training Data

---

### Official Review · AnonReviewer4 · 2020-10-27
**Excellent paper with nice insight on using gradient embeddings to improve private learning**

**Rating:** 9
**Confidence:** 4

**Review:**

This paper proposes a simple but highly effective approach to improve differentially private deep learning, by projecting gradients to a small subspace spanned by public data.
This idea has been proposed before, but this paper seems to have the unique (and important) insight to release noisy residual gradients, so as to ensure that the final gradient estimation remains unbiased.
Experiments on MNIST, SVHN and CIFAR-10 show that the proposed approach significantly improves the privacy-utility tradeoff of DP-SGD.

This paper is excellent. It is well motivated, as the assumption of access to a small amount of public data seems very reasonable in practice. The proposed algorithm is quite natural. The key insight is to release private noisy estimates of both the low-dimensional gradient embedding *and* the residual (low-norm) gradient. This is easily shown to produce an unbiased estimate of the gradient, which is crucial for performance.
The experiments convincingly show that this approach leads to strong performance improvements, and I predict that this simple technique will be applied in many subsequent works.

I have a number of (mostly minor) comments and suggestions for improvement:

- In Algorithm 1, and in the main text, it would be good to clarify the clipping applied to both the embedded and residual gradients. It is also worth noting that in practice, the estimated gradient is probably *not* quite unbiased because of this clipping (the same holds for standard DP-SGD of course).

- You mention that you use "privacy amplification by subsampling" to compute the DP guarantees. It would be good to clarify this, since some of these theorems apply specifically to a "subsample + gaussian mechanism" approach, which your algorithm doesn't quite match (GEP applies two independent Gaussian mechanisms to functions of a subsampled gradient).
This seems like it could be resolved as follows:
- you have embedded gradients Wi clipped to norm C1
- you have residual gradients Ri clipped to norm C2
- You can "stack" these as gradients gi = [Wi/C1 ri/C2] of norm S=sqrt(2), and then apply a single Gaussian mechanism with sensitivity S to this stacked gradient. After that, you can rescale the two components by C1 and C2 respectively. This seems identical to your proposed approach of applying two Gaussian mechanisms with sensitivities scaled by a factor \sqrt(2).


- Only in Section 4 do you mention that you split the gradient into different groups to fit them in memory. It would be nice to see this discussed earlier on in the paper. E.g., the abstract claims that GEP has "low computational cost", which then seems strange when Section 3.1 says that the cost is O(m*k*p), which would be huge even for moderate DNNs. Section 3.2. mentions using a ResNet-20 with k=2000 basis vectors, which wouldn't fit in any GPU memory if it weren't for the grouping. While reading these sections, I was confused about how this could possibly be implemented at a low cost, as described.
Following on this, it would be nice to have some results on the additional computational and memory costs of GEP in the evaluation.

- Papernot et al. show that the choice of architecture can have a large influence on DP-SGD (https://arxiv.org/abs/2007.14191). The results they obtain are sometimes better than your GP baseline. It would be interesting to see if you can get even better results with GEP with such architectures.

- This is somewhat orthogonal to your approach, but do you have a sense of whether 2000 public ImageNet samples could be leveraged in other ways than as gradient basis vectors? E.g., on CIFAR-10, it is well-known that unsupervised dictionaries of data patches can achieve >80% accuracy. So one could also consider using the public data to learn a patch dictionary, and then simply train a small private classifier on top of these features. Maybe this would perform better than GEP?

---

> ### Author Response · Authors · 2020-11-17
> **Response to Reviewer 4**
>
> Thanks for your detailed reading. We greatly appreciate your supportive and valuable comments.
>
> · Q: It would be good to clarify the clipping applied to both the embedded and residual gradients. && It is worth noting that in practice the estimated gradient is probably not quite unbiased because of clipping.
>
> A: We add comments about clipping in both Algorithm 1 and the main text (in Remark 1 and the afterward discussion). We clarify that the clipping could hurt the unbiased property in theory and practice. In the literature, it is common to choose a large clipping threshold to make the gradient unbiased so that the theoretical analysis is easy to go through.
>
> · Q: Clarification on the use of "privacy amplification by subsampling" is needed.
>
> A: Thanks for pointing this out. The output of GEP is exactly the same as the output of your example. Hence the privacy loss of GEP can be computed by the numerical tool in [1]. In our implementation, we double the privacy loss of standard “subsample + gaussian mechanism” as the privacy loss of GEP.   While the privacy loss is computed accurately, it is more rigorous to understand the privacy loss of GEP by your example. We adopt your suggestion in Section 4 of the updated submission.
>
> · Q: The discussion on “splitting the gradient into different groups” should appear earlier in the paper.
>
> A: In the updated submission, we introduce “splitting parameters into different groups” as another important technique in Section 3.1. The computational cost of each power iteration is O($mkp/g$) if the parameters are divided into $g$ groups evenly. We also add a table in Appendix B to show the additional computational and memory costs of GEP.
>
> · Q: It would be interesting to see whether GEP can get better results with the techniques in [2].
>
> A: We have tried the tempered sigmoid activation in [2]. We switch the activation function of ResNet20 from ReLU to Tanh. In this case, the best non-private test accuracy we can get on CIFAR10 is ~72%. Therefore, we still use ReLU as the activation function in Section 4. Another technique in the experiments of [2] is using simpler architecture for training DP models.  We apply GEP on models with fewer parameters. On a ResNet14 model, GEP achieves 75.3% accuracy on CIFAR10, which is 0.4% higher than the result in Section 4. This is an interesting direction for future work.
>
> · Q: Can ImageNet samples be leveraged in other ways than as gradient basis vector?
>
> A: We thank the reviewer’s thought-provoking question. We differentially privately train a linear classifier using the features produced by a model pre-trained on unlabeled ImageNet using the setting in [3]. The implementation details and experiment results are presented in Appendix A of the revised version. The features produced by pretrained model can indeed greatly improve the utility with differential privacy guarantee. We further examine whether GEP can improve the performance of such private linear classifiers. GEP consistently outperforms conventional gradient perturbation under various privacy budgets. For example, with privacy bound $\epsilon=2$, GEP achieves *94.8%* accuracy on CIFAR10, improving over the GP baseline by 1.4%.
>
> [1]: Renyi Differential Privacy of the Sampled Gaussian Mechanism, Arxiv 2019
>
> [2]: Tempered Sigmoid Activations for Deep Learning with Differential Privacy, Arxiv 2020
>
> [3]: A Simple Framework for Contrastive Learning of Visual Representations, Arxiv 2020

---

> > ### Comment · AnonReviewer4 · 2020-11-20
> > **On the transfer learning experiments**
> >
> > Thanks for making all these changes. This improves an already excellent paper.
> >
> > Regarding the new results with transfer learning I have two comments/questions:
> >
> > 1) Isn't it surprising that GEP helps even in such low-dimensional settings? (the SIMCLR representations have dimension 4096 if I'm not mistaken).  Presumably this means that even in this setting, the gradients have low stable rank?
> >
> > 2) In my original comment about other ways of leveraging public data, I was thinking more about low public data regimes. That is, could you use the 1000-2000 public data points for something additional than a gradient basis? Presumably, training on such a small amount of data is not sufficient for learning a good representation as in SIMCLR, but it might still help to "initialize" a model for private training with GEP.

---

> > > ### Author Response · Authors · 2020-11-21
> > > **Response to new comments**
> > >
> > > Thanks again for your constructive suggestions/comments that help us to improve this submission.
> > >
> > > 1.	The gradient matrix in this setting does have a low stable rank. The linear model has a dimension of 40960, which is a 4096$\times$10 matrix that maps the representation into probabilities for 10 classes. The stable rank of the gradient matrix at initialization is only 12.7.
> > >
> > > 2.	A small amount of public data is indeed not sufficient for unsupervised learning methods such as SimCLR to learn a good representation. We have tried a simple pre-training method using only 1000-2000 public data points at the early stage of our experiments. We use those public data points (with correct labels) to pre-train the model before private learning with GEP. It turns out that this method has higher projection error and worse test performance. We find the reason is the model overfits public data points and hence the anchor subspace contains less information about the private gradients.  Nonetheless, finding new ways to leverage the auxiliary data is definitely an interesting and important problem and we are still actively thinking about it.

---

### Official Review · AnonReviewer1 · 2020-10-28

**Rating:** 7
**Confidence:** 3

**Review:**

Paper proposes gradient embedding as a potential solution to boost the privacy-utility trade-off in differentially private machine learning via gradient perturbation. Main idea is to project the gradients into a lower-dimensional subspace before adding noise, so that the curse of dimensionality in DPSGD can be somewhat dealt with.

I like the paper and the motivation as high-dimensionality can be considered as a major driver behind the privacy-utility trade-off in DPSGD. Also, the requirement for a public data is much more acceptable in the provided scenario, where it is only required to get anchor gradients. Overall empirical evaluation is adequate, however, it would have been nice to see PATE's numbers on all comparisons, especially given that the implementation is publicly available. I would have also liked the source-code as these type of claims are best verified empirically, especially when we are using gradient projection that can can lead to projection errors and the page limit in the submission is nearly not enough for a thorough evaluation.

---

> ### Author Response · Authors · 2020-11-17
> **Response to Reviewer 1**
>
> Thank you for your supportive review. Here is our response to your concerns.
>
> ·  Q：PATE’s numbers on all comparisons are needed.
>
> A：We are still actively working on getting more experiment results of PATE and will update the submission once they are finished. The current difficulty is that the PATE code is based on an outdated version of Tensorflow and we need time to adapt it into friendly environment.
>
> From our experience, PATE may not be good at complex datasets such as CIFAR10 because it divides the dataset into hundreds of shards to train hundreds of teacher models. The performance of each teacher model is bad because of the extreme small size of training data, which could further limit the performance of the student model.
>
> ·  Q：The source code is not publicly available.
>
> A：Thanks for the suggestion. We have uploaded our source code with a readme file in the updated submission.

---

> > ### Author Response · Authors · 2020-11-20
> > **PATE's numbers on all comparisons are now available**
> >
> > We take some effort to successfully train PATE on all the settings in Table 1. The results are updated in the submission draft. The experiments use the official implementation \[1\] and follow the experimental setup in [2,3]. Here is a comment on PATE’s results.
> >
> > PATE has the best performance on MNIST but is not so good on SVHN and CIFAR10. We speculate that this is because of the following reasons. PATE needs to divide the training data into a large number of disjoint subsets (this setting naturally arises in some specific scenarios such as when the data belong to different institutions). Each teacher model only has access to one shard.  Although such division has relatively small impact on performance for simple dataset (MNIST), the limited size of training data for each teacher model could become bottleneck of the performance for complex datasets (CIFAR).
> >
> > In comparison, the GEP algorithm performs consistently well irrespective of the datasets and outperforms PATE by a large margin on complex datasets. Moreover, GEP achieves larger improvement than PATE as increasing the privacy budget, which could be due to that PATE suffers the bottleneck of limited training data for each teacher model.
> >
> > \[1\]:  https://github.com/tensorflow/privacy/tree/master/research
> >
> > [2]: Semi-supervised Knowledge Transfer for Deep Learning from Private Training Data, ICLR 2017
> >
> > [3]: Scalable Private Learning with PATE, ICLR 2018

---

> > > ### Comment · AnonReviewer1 · 2020-11-22
> > > **response**
> > >
> > > Thank you for making the changes. It makes the paper much stronger. I have increased my score accordingly.

---

> > > > ### Author Response · Authors · 2020-11-23
> > > > **Thank you for your acknowledgement of our work!**
> > > >
> > > > We are happy to know that the reviewer is more satisfied with our work. Thank you for your time.

---

### Official Review · AnonReviewer2 · 2020-10-30
**Review for "Do not Let Privacy Overbill Utility: Gradient Embedding Perturbation for Private Learning"**

**Rating:** 6
**Confidence:** 4

**Review:**

Summary: The paper focuses on training deep learning models under differential privacy. The main contribution is a technique based on  projecting the gradient into a non-sensitive anchor subspace and then perturbing the projected gradient (and the residual).

Comments:

It is well-known that a straightforward way of perturbing the gradients for achieving differential privacy requires that the noise magnitude has to scale with the dimensionality of the gradients.

Now to reduce the dependence on dimension the authors use a simple idea: reduce the dimension of the gradients using an embedding and then add noise into the low-dimensional gradients. They assume the presence of non-sensitive data for this purpose. The rough outline is as follows: 1) use the non-sensitive data to compute a subspace using a principal components approach, 2) project the private gradient into this subspace to obtain a low-dimensional gradient embedding and the residual, 3) perturb the gradient embedding and residual gradient separately for DP purposes, d) use the perturbed gradients in a DP-GD iteration.

Strengths:
1)	The problem of reducing noise in differentially private ML is an important practical problem.
2)	The experiments shows benefit of this approach over Abadi et al.
I think the most interesting part of the paper is the experimental evaluation. There is no strong theoretical component in the paper.

Here are some questions/suggestions:
1) Can you provide convergence bounds achieved with using gradients produced by GEP (Algorithm 1) in standard convex/nonconvex optimization settings.
2)  Are there any good bounds on S_1 and S_2?
3) I did not understand the role of Lemma 3.1? Don’t you require the anchor subspace to be non-sensitive (not depend on sensitive gradients)?
4) How crucial is the choice of picking the right auxiliary dataset. Providing experimental results with different choices of the auxiliary dataset might help.
5) Why does not the random projection using JL work? The authors allude to an empirical observation in the text which I did not find.

---

> ### Author Response · Authors · 2020-11-17
> **Response to Reviewer 2**
>
> Thank you for your review and valuable comments. Our point-to-point responses are  listed below.
>
> 1） Q：Can you provide convergence bounds achieved by using gradients produced by GEP (Algorithm 1) in standard convex/nonconvex optimization settings?
>
> A：As the reviewer suggested, in the revised version, we give an expected excess error bound achieved with GEP in standard convex setting (Theorem 3.3 in Section 3.3).   We did not include the convergence bounds in the initial submission because they require strong assumptions, i.e.,  convexity,  exact low-rankness of the gradient matrix that the deep neural network objective does not satisfy.
>
> There are two terms on the right hand side: one is $\frac{k}{n\epsilon}$, where $k$ is the dimension of anchor subspace and the other is the previous best utility bound multiplied by a term representing “the average projection error over the training process”. When the gradient matrices are low-rank over the training procedure, the utility bound does not depend on the ambient dimension but only depends on $k$ by properly choosing $m$ and $k$.  In general case, our bound justifies the superiority of GEP because of Lemma 3.2 and the empirical observation. We note that our bound is at least as good as previous best bound under the same assumptions.
>
> 2） Q：Are there any good bounds on S_1 and S_2?
>
> A：A naïve bound for both $S_1$ and $S_2$ is the norm of the original gradients or the Lipschtiz constant. In practice, we use Lipschitz constant/gradient norm to bound the $S_1$. For $S_2$, we use Lemma 3.2 to give a bound on the expected residual gradient norm, which could be as small as $\sum_{k’>k}\lambda_{k’}(\Sigma)$, where $\lambda_{k’}(\Sigma)$ is the $k’$-th largest eigenvalue of the population gradient covariance matrix (uncentered). This motivates us to choose a small pre-defined threshold $S_2$ in practice.
>
> 3）Q：Lemma 3.1 assumes the anchor gradients are sampled from sensitive gradients.
>
> A：The original Lemma 3.1 was to give a hint on how large the residual gradients could be. We thank the reviewer for pointing out the confusing assumption in original Lemma 3.1. We update this part with a new Lemma 3.2 that measures the residual gradient norm under the same setting as in GEP algorithm.
>
> 4）Q：How crucial is the choice of picking the right auxiliary dataset?
>
> A：We conduct experiments with different choices of auxiliary datasets to examine this. The results suggest that  different auxiliary datasets have very small impact on the test accuracy.  For the CIFAR10 classification task with the same DP guarantee, GEP achieves 75.1%/74.7%/74.8% test accuracy when the auxiliary data are sampled from part of CIFAR10 test data/CIFAR100/ImageNet, respectively. The details can be found in Appendix C of the updated submission.
>
> 5）Q：Why does not the random projection work?
>
> A：This is because random projection can only preserve the pairwise distance but not guarantee accurate gradient reconstruction. In the gradient descent algorithm, the accurate gradient direction (gradient reconstruction) is important to guarantee the convergence. In the updated version, we add two plots in Appendix C to show the projection errors of random basis vectors on both CIFAR10 and SVHN datasets are very high.

---

> > ### Comment · AnonReviewer2 · 2020-11-23
> > **Reply to author responses**
> >
> > Thanks for your responses. Why would be gradients be in a low-dimensional subspace? Even in sparse optimization, gradients are not sparse.  Can you explain a natural optimization problem where the gradients are always in a low-dimensional subspace?

---

> > > ### Author Response · Authors · 2020-11-23
> > > **Response to new question**
> > >
> > > Thank you for the question.
> > >
> > > Take linear regression as an example. The gradients will stay in the low-dimensional subspace spanned by the feature space of the data matrix. Let $X\in \mathbb{R}^{n\times p}$ with $n$ samples and $p$ features be the data matrix, $y \in \mathbb{R}^{n\times 1}$ be the response vector, $\theta \in \mathbb{R}^{p\times 1}$ be the model parameter, and $\ell= \frac{1}{2}\left\lVert X\theta - y \right\rVert^2$ be the loss function. The individual gradients of $\ell(\theta, x_i,y_i)$ with respect to $\theta$ are $$x_i(x_i^T\theta- y_i)$$, always stay in the subspace spanned by the columns of $X$. Hence, the gradient matrix would be low-rank (not necessarily sparse) if data matrix $X$ is low-rank. For real problems, it is reasonable to assume that the data matrix is low-rank because of the correlation between features. For example, there exist high correlations between neighboring pixels in natural images. Furthermore, it has also been observed that the redundancy in features leads to approximately low-rank gradients [1, 2] for deep learning models, which is enough to show the advantage of GEP.
> > >
> > > [1]: Gradient Descent Happens in A Tiny Subspace, Arxiv 2018
> > >
> > > [2]: PowerSGD: Practical Low-Rank Gradient Compression for Distributed Optimization, NeurIPS 2019

---

### Author Response · Authors · 2020-11-17
**General Response**

We thank all the reviewers for the constructive comments. According to reviewer’s suggestion, we have updated our submission to include the following changes:

1）In Section 3.3, we add an expected excess error bound achieved with GEP in standard convex setting and compare with existing results.

2）In Appendix A, we add experiments of training differentially private models based on the pre-trained models and demonstrate the benefit of GEP over standard DPSGD.

3）In Appendix C, we add some ablation study to examine: 1) the influence of choosing different auxiliary datasets; 2) the influence of using different numbers of anchor gradients; 3) the projection error of random basis vectors.

4）We upload our source code with a readme file.

---

### Decision · Program_Chairs · 2021-01-07
**Final Decision**

**Decision:**

Accept (Poster)

**Comment:**

The paper introduces a method for differentially private deep learning, which the authors term Gradient Embedding Perturbation. This is similar to several (roughly) concurrent works, which project gradients to a subspace based on some auxiliary public data. However, a crucial difference involves the use of the residual gradients, which allows the method to achieve the first significant accuracy gains using subspace projection. The reviewers believe this method will be important for the practice of DP deep learning.